# Knee Laxity in the Menstrual Cycle after Anterior Cruciate Ligament Reconstruction: A Case Series

**DOI:** 10.3390/ijerph20032277

**Published:** 2023-01-27

**Authors:** Mayuu Shagawa, Sae Maruyama, Chie Sekine, Hirotake Yokota, Ryo Hirabayashi, Ryoya Togashi, Yuki Yamada, Haruki Osanami, Daisuke Sato, Mutsuaki Edama

**Affiliations:** 1Institute for Human Movement and Medical Sciences, Niigata University of Health and Welfare, 1398 Shimami-cho, Kita-ku, Niigata 950-3198, Japan; 2Albirex Niigata Inc. 5-1923-23, Higashiko, Seiro-machi, Kitakanbara-gun, Niigata 957-0101, Japan

**Keywords:** estradiol, anterior knee laxity, genu recurvatum, muscle stiffness

## Abstract

The aim of this study was to compare anterior knee laxity (AKL), genu recurvatum (GR), and muscle stiffness between reconstructed and contralateral sides in females who underwent anterior cruciate ligament (ACL) reconstruction during early follicular and ovulatory phases. AKL was measured as an anterior displacement of the tibia using a KS measure. GR was measured as the range of motion of knee hyperextension using a hyperextension apparatus. Muscle stiffness was measured for semitendinosus (ST) and biceps femoris long head (BF) using a MyotonPRO. The study investigated eighteen knees in nine females (Age, 20.4 ± 1.5 years; BMI, 21.5 ± 1.5) with normal menstrual cycles at least 1 year after reconstruction using hamstring autograft. E2 (Estradiol) concentration did not differ between the two phases, but AKL on the reconstructed side was lower during the ovulatory phase (8.3 [5.9–9.3] mm) than during the early follicular phase (9.4 [7.3–9.7] mm) (*p* = 0.044, *r* = 0.756), whereas there was no significant difference between the two phases on the contralateral side. AKL side-to-side difference, GR, and muscle stiffness (ST and BF) on both sides did not differ in either phase. These results indicate that AKL may behave differently on the reconstructed and contralateral sides during the menstrual cycle.

## 1. Introduction

Female athletes reportedly show a higher incidence of anterior cruciate ligament (ACL) injuries, up to 2.8–3.5 greater than male athletes [1,2] and ACL injuries occur more frequently in the early follicular and ovulatory phases than in the luteal phase of the menstrual cycle [3]. This suggests that fluctuations in female hormones contribute to the occurrence of ACL injury [4]. ACL injury is multifactorial [5], with one risk factor being joint laxity such as anterior knee laxity (AKL) [6] and genu recurvatum (GR) [7]. Variations in joint laxity in healthy knees have been observed over the menstrual cycle [8,9]. A systematic review concluded that greater AKL during the ovulatory phase may increase the risk of ACL injury, but further research is needed [3]. Maruyama et al. observed that AKL increased during the ovulatory phase in participants with GR [10]. The magnitude and timing of the effect of female hormones on joint laxity in the healthy knee may thus differ among individuals. To the best of our knowledge, earlier research only involved healthy knees and the effect of the menstrual cycle on joint laxity in reconstructed knees has not been evaluated. The risk of secondary injuries, such as re-injury and contralateral injury, is four times higher in females than in males [11], so prevention of ACL injury needs to be considered a pressing issue.

Ligamentization of the reconstructed ligament is completed approximately 1 year after surgery, replacing almost all of the native tissue structure of the ACL [12]. The International Documentation Knee Committee’s assessment of AKL side-to-side difference [13] is commonly used in clinical settings as an outcome measure after ACL reconstruction [14]. However, fluctuations in joint laxity on the reconstructed side during the menstrual cycle have not been clarified and differences in joint laxity between reconstructed and contralateral sides could lead to cyclic fluctuations in side-to-side differences in AKL. To provide appropriate treatment and rehabilitation after re-injury or contralateral injury, the effect of the menstrual cycle on the reconstructed knee needs to be clarified.

Estradiol (E2) is an ovarian hormone and one of the estrogens, with levels fluctuating throughout the menstrual cycle. E2 has been considered a cause of AKL fluctuations in healthy knees [15]. Joint laxity is restricted by a combination of ligaments and surrounding tissue, such as muscle. E2 receptors are expressed in the ACL [16], fascia [17], muscle tissue [18], and tendon [19]. Yu et al. reported that increased E2 concentrations in human ACL cells resulted in decreased type I collagen synthesis and fibroblasts [20]. The density of cross-links between collagen also affects the stiffness of ligaments along with the amount of collagen [21]. Lee et al. demonstrated that artificial ligaments with different amounts of added estrogen showed no change in collagen content but a significant decrease in lysyl oxidase, a collagen cross-linking enzyme, which correlated with a decrease in the mechanical stiffness of ligaments [22]. E2 might therefore affect the stiffness of ligaments and the elasticity of muscles and fascia and is thought to be the cause of variations in joint laxity during the menstrual cycle. Previous studies that examined joint laxity changes during the menstrual cycle did not simultaneously look at muscle stiffness and were unable to identify the tissues responsible for changes in joint laxity [8,9,10]. Therefore, the observation of muscle stiffness as well as evaluation of joint laxity is necessary.

The purpose of this study was to compare joint laxity (AKL and GR) and muscle stiffness between the reconstructed and contralateral sides during the early follicular and ovulatory phases in females who underwent ACL reconstruction. The hypothesis was that differences would be seen in joint laxity (AKL and GR) behavior and muscle stiffness between the reconstructed and contralateral sides. The contralateral side would have greater joint laxity and lower muscle stiffness than the reconstructed side in the ovulatory phase.

## 2. Materials and Methods

### 2.1. Subjects

An initial 97 female students at Niigata University of Health and Welfare answered a questionnaire that asked about the history of ACL injury and menstrual condition. Inclusion criteria for this study were subjects who had undergone primary unilateral ACL reconstruction using semitendinosus (ST) autograft at least 1 year earlier, who had not used oral hormone contraceptives in 6 months, and who had a cycle length of more than 25 and less than 38 days. The level and experience of sports that they joined were not restricted. Participants who have no history of ACL injury, are using hormone contraceptives, have amenorrhea, are conflicted for scheduling, declined participation, had undergone ACL reconstruction using patella tendon graft, and had secondary contralateral ACL reconstruction were excluded. In subjects who met all criteria and agreed to participate in the experiment, the subjective knee function was assessed by International Documentation Committee (IKDC), subjective knee evaluation form was translated into Japanese by American Orthopaedic Society for Sports Medicine [23], and activity level assessed using Tegner Activity Scale before and after surgery [24]. The study was conducted in accordance with the Declaration of Helsinki after being approved by the ethics committee at our institution (approval No. 18671). The study information was thoroughly presented to the subjects, and all subjects submitted written, informed consent before participating in the study.

### 2.2. Recording the Menstrual Cycle

Subjects were told to use a basal body thermometer (CTEB503L electronic thermometer; Citizen Systems Co., Tokyo, Japan) to assess basal body temperature every morning. An ovulation prediction kit (Doctor’s Choice One Step Ovulation Test Clear; Beauty and Health Research, Torrance, CA, USA) was used to estimate the day of ovulation, starting the day after the end of period bleeding, and continuing until a positive result was obtained. Subjects were asked to use the ONE TAP SPORTS athlete condition management system (Euphoria Co., Tokyo, Japan) to record the start and end of the period, basal body temperature, and ovulation prediction kit results.

### 2.3. Timing of Measurement

Joint laxity (AKL and GR) and muscle stiffness were measured for 2 days during the early follicular and ovulatory phases. The onset of menstruation was considered the first day of the cycle and the early follicular phase was defined for 2 days in a row, from the second to the fourth day of the cycle. The ovulatory phase was considered 2 days in a row, from the second to the fourth day following a positive result from the ovulation prediction kit. A saliva sample was collected for E2 concentration analysis only on the first day of assessment for each phase. Fluctuations in joint laxity and muscle stiffness were taken into account throughout the day; all measures were performed between 07:00 a.m. and 12:00 p.m. The experiment room was set to a temperature range of 20–25 °C.

### 2.4. Measurement Methods

#### 2.4.1. Estradiol Concentration

Using a saliva collection kit (SalivaBio A; Salimetrics, Carlsbad, CA, USA), E2 concentration was collected according to the handbook [25]. To avoid any influences on E2 concentration, participants obeyed the following instructions recommended by the manufacturer before saliva collection: (1) no food intake within 60 min; (2) no dairy products within 20 min; (3) no alcohol within 12 h; (4) no sugary, acidic, or caffeinated drinks within 20 min; and (5) no saliva collection performed during 48 h after dental treatment. In addition, participants were instructed to rinse their mouths before the experiment began to remove any food particles. Saliva samples were taken more than 10 min after mouth-rinsing to prevent decreases in E2 concentrations. A straw (Saliva Collection Aid; Salimetrics, Carlsbad, CA, USA) was used to drain saliva into a saliva collection container (Cryovial; Salimetrics, Carlsbad, CA, USA). Immediately following collection, the saliva sample was frozen in a freezer at below −80°C. After gathering all samples, Funakoshi Corporation (Tokyo, Japan) was entrusted with the analysis of E2 concentrations. The High Sensitivity Salivary 17-Estradiol Enzyme Immunoassay Kit (Salimetrics, Carlsbad, CA, USA) was used to measure E2 concentrations. Samples were thawed at room temperature, mixed via vertexing, centrifuged at 1500× *g* for 15 min, then analyzed using an enzyme-linked immunosorbent assay. Dilutions were all consistently onefold (undiluted solution).

#### 2.4.2. Anterior Knee Laxity

Based on prior work [10], AKL was calculated as the anterior tibial displacement of the femur following application of 133 N of anterior stress on the tibia. According to the measuring criteria of Nippon Sigmax Corporation (Tokyo, Japan), measurements were carried out using the KS Measure (KS Measure KSM-100; Nippon Sigmax Co., Ltd., Tokyo, Japan). The subject was placed in a supine position with the knee rest under the distal posterior aspect of the thigh and the footrest beneath the foot, with the knee joint flexion set at around 30° using a goniometer (Goniometer; Nishikawashinwa, Tokyo, Japan). The KS measure was positioned to be perpendicular to the tibia. Both the reconstructed and contralateral sides were measured in random order. Five measurements were conducted for each knee, then the highest and lowest data points were discarded, and the three remaining measurements were averaged.

#### 2.4.3. Genu Recurvatum

To measure the range of motion of knee extension, GR was evaluated using a hyperextension apparatus (Takei Scientific Instrument Co., Niigata, Japan). This was the same apparatus used in prior work (Figure 1) under the same methods [26]. Subjects sat on the hyperextension apparatus in a relaxed position, lying against the backrest cushion. When the height of the seat and height of the foot scales were equal, knee joint extension was considered to be 0°. To obtain 0° of hip adduction, the lower limb positions on the right and left were adjusted, and the length from the knee to the heel was varied by setting the heel on the footrest. A knee-fixation belt was used to fix the proximal patella of the femur. The knee joint was extended by moving the foot elevation screw upward at a rate of less than 1°/s until a signal was provided by the subject. Subjects were instructed to indicate as soon as they felt any discomfort or pain in the knee joint. Extended knee position was maintained for 10 s after the participant signaled, and the measurement of the increase in foot height was recorded using a digital camera (Power Shot SX 740 HS; Canon, Tokyo, Japan). Five measurements were made after one trial test. The highest and lowest data points were discarded, and the three remaining measurements were averaged. The five trials were separated via a 10 s rest. Both reconstructed and contralateral sides were measured in random order. Length from the center of the knee to the heel (L) and the increase in value in foot height (H) were used to calculate the maximum angle of passive knee hyperextension. The equation is as follows:GR [°] = arctan(H/L) * 57.2958

#### 2.4.4. Muscle Stiffness

Muscle Stiffness of the ST and biceps femoris long head (BF) were measured using a MyotonPRO digital palpation device (Myoton AS, Tallinn, Estonia), a device that can non-invasively evaluate the mechanical properties of muscle on the surface of the skin. The subject lay prone in position on a bed with the foot hanging from the edge of the bed. Measurement points were identified by being displayed on a diagnostic ultrasound imaging system (Aplio500; Canon Medical Systems Corporation, Tochigi, Japan) at the midpoint between the greater trochanter of the femur and the head of the fibula and marked on the skin surface above the muscle belly based on previous research [27]. After marking, the subject took 10 min to rest on the bed to relax. The measurement technique used in the MyotonPRO is based upon the application of 5 mechanical impulses with a brief duration (time, 15 msec; force, 0.4 N) under steady pre-compression force (0.18 N) of the subcutaneous tissue layer above the muscle being evaluated [28]. A device probe (diameter, 3 mm) set perpendicular to the surface of the skin delivers mechanical deformation. Damped oscillation produced by the muscle in response to a brief mechanical impulse is detected using an acceleration sensor connected to the frictionless measurement mechanism of the device. The order of measurement was random for limbs and muscles. The highest and lowest data points of five measurements were discarded and the three remaining measurements were averaged for each muscle.

### 2.5. Reliability of Measurements

#### 2.5.1. Test–Retest Reliability

Eight adult males (sixteen legs) (mean [±standard deviation] age, 19.4 ± 0.8 years; height, 171.2 ± 5.5 cm; weight, 64.5 ± 9.5 kg) were measured for joint laxity (AKL and GR) and muscle stiffness (ST and BF). To avoid the influence of hormone fluctuation, males were chosen instead of females to evaluate the reliability of measurements. After more than 2 days but within less than 7 days, the same examiner retested the methods of measurement using the same procedures as in the actual experiment. Inter-class correlation coefficients (ICC (1, 3)) were then determined.

#### 2.5.2. Interrater Reliability

GR and muscle stiffness (ST and BF) were evaluated by 2 examiners on the same day. GR was measured by the first experimenter, then by the second experimenter 30 min later. Muscle stiffness was measured after 10 min. Intra-class correlation coefficients (ICC ((2, 3)) were calculated. According to the criteria of Landis et al. [29], reliability is considered “almost perfect” for ICCs of 0.81 or more, which is used to evaluate reliability.

### 2.6. Statistical Analysis

Data were analyzed using IBM SPSS Statistics version 28.0 (IBM Corp., Armonk, NY, USA). A Shapiro–Wilk test determined that our data followed a parametric distribution except for AKL side-to-side difference in the early follicular phase; however, a non-parametric approach was used due to the small sample size to assess its true distribution. A Wilcoxon signed-rank test determined whether there was a statistically significant difference between two phases (early follicular and ovulatory) of AKL, GR, and muscle stiffness (ST and BF) on the reconstructed and contralateral sides, and AKL side-to-side difference, E2 concentration. Also, comparison of reconstructed and contralateral sides of AKL, GR, and muscle stiffness (ST and BF) in each phase were analyzed using a Wilcoxon signed-rank test. Values of *p* < 0.05 were considered statistically significant. Based on the Wilcoxon signed-rank test, effect size was calculated using Rank Biserial Correlation (r) [30] and categorized as small (0.1), medium (0.3), or large (0.5) [31].

## 3. Results

Subjects in this study were nine females (eighteen legs) who met all criteria and agreed to participate in the experiment (Figure 2). Table 1 shows the demographic data, IKDC subjective score, and activity level of the subject. Results for E2 concentration, AKL, side-to-side difference, GR, and muscle stiffness (ST and BF) are shown in Table 2, Table 3 and Table 4. E2 concentration, AKL, side-to-side difference, GR, and muscle stiffness (ST and BF) are shown as the median (interquartile range).

### 3.1. Comparison of the Early Follicular and Ovulatory Phases

E2 concentration did not differ significantly between the early follicular (1.2 [0.9–1.6] mm) and ovulatory phases (1.5 [1.1–1.8] mm) (*p* = 0.374, *r* = 0.333) (Table 2). AKL on the reconstructed side was lower during the ovulatory phase (8.3 [5.9–9.3] mm) than during the early follicular phase (9.4 [7.3–9.7] mm) (*p* = 0.044, *r* = 0.756), but not on the contralateral side (early follicular: 5.3 [4.7–6.9] mm, ovulatory: 4.8 [4.4–7.5] mm) (*p* = 0.515, *r* = 0.244) (Table 3). AKL side-to-side differences did not differ significantly between phases (early follicular: 3.3 [2.9–4.4] mm, ovulatory: 3.5 [1.0–4.0] mm) (*p* = 0.051, *r* = 0.733) (Table 2). GR, ST stiffness, and BF stiffness did not differ significantly during the two phases in both limbs (Table 3).

### 3.2. Comparison of Reconstructed and Contralateral Sides

AKL was greater on the reconstructed side than on the contralateral side in both phases, the early follicular (reconstructed: 9.4 [7.3–9.7] mm, contralateral: 5.3 [4.7–6.9] mm) (*p* < 0.008, *r* = 1.000) and the ovulatory (reconstructed: 8.3 [5.9–9.3] mm, contralateral: 4.8 [4.4–7.5] mm) (*p* = 0.021, *r* = 0.867). (Table 4). GR was greater on the contralateral side than on the reconstructed side in the two phases, the early follicular (reconstructed: 7.7 [5.6–9.1] °, contralateral: 9.7 [7.1–12.0] °) (*p* = 0.021, *r* = 0.867) and the ovulatory (reconstructed: 7.5 [6.4–9.9] °, contralateral: 10.2 [7.9–12.0] °) (*p* = 0.028, *r* = 0.822). (Table 4). ST stiffness and BF stiffness in both side limbs did not differ significantly between the early follicular and ovulatory phases (Table 4).

### 3.3. Reliability of Measurements

ICC (1,3) for AKL was 0.81. ICC (1,3) and ICC (2,3) for GR were 0.941 and 0.937, respectively. ICC (1,3) and ICC (2,3) for ST stiffness were 0.970 and 0.980, respectively. ICC (1,3) and ICC (2,3) for BF stiffness were 0.934 and 0.974, respectively. All reliability of the measurements obtained an “almost perfect” according to Landis et al. [29]. 

## 4. Discussion

In the present study, joint laxity (AKL and GR) and muscle stiffness (ST and BF) on the reconstructed and contralateral sides during the menstrual cycle were compared in women who had undergone ACL reconstruction. This is the first study to examine joint laxity on the reconstructed side during the menstrual cycle. There were four main findings. First, AKL on the reconstructed and contralateral sides showed different fluctuations during the cycle (AKL on the reconstructed side was lower in the ovulatory phase than in the early follicular phase, and AKL on the contralateral side did not differ significantly between the early follicular and ovulatory phases). Second, AKL side-to-side difference did not differ significantly during the menstrual cycle. Third, joint laxity (AKL and GR) on the reconstructed and constructed sides showed significant differences. Fourth, muscle stiffness did not change between phases of the menstrual cycle.

AKL on the reconstructed side was lower in the ovulatory phase than in the follicular phase. Grafts of reconstructed ligaments reportedly undergo remodeling to resemble the native ACL at 1 year [12], whereas another study reported that remodeling continued at 9 years postoperatively [32]. This reconstructed ligament may respond differently to E2 than the native ACL, since E2 promotes type I collagen synthesis [33] and may have a predominant effect in assisting remodeling in the reconstructed ligament. However, since E2 concentration did not differ between the early follicular and ovulatory phases in this study, progesterone and relaxin, as female hormones other than E2, may also exert influences on the reconstructed ligaments. Further studies are needed because the effects of hormones on reconstructed ligaments have yet to be clarified.

In the present study, AKL on the contralateral side did not differ significantly between the early follicular and ovulatory phases. A systematic review has stated that AKL fluctuation in healthy knees during the menstrual cycle is higher during the ovulatory phase [3]. In contrast, Park et al. found that AKL in 10 of 26 healthy women was maximal during the early follicular phase [34]. Accordingly, individual differences may exist in the variability of joint laxity. Furthermore, AKL has been reported to fluctuate during the cycle among individuals with GR [10], with some people being laxity responders to the menstrual cycle and others being non-responders [35], suggesting that certain populations may be more susceptible to hormonal influences during the menstrual cycle. Among individuals with a history of ACL injury, however, joint laxity on the contralateral side did not change between the early follicular and ovulatory phases, indicating that this population is not susceptible to the impact of female hormones.

Considering the clinical meaning of the finding that there was no significant change in AKL side-to-side difference between the early follicular and ovulatory phases, the menstrual cycle may not need to be considered when evaluating the side-to-side difference. Continuing to measure outcomes at a defined phase in the menstrual cycle is difficult because the menstrual cycle is not able to control, and patients would not come to a clinician at the same phase. For this reason, AKL side-to-side differences have not been measured with a specific phase in the menstrual cycle. If the side-to-side difference changes during the menstrual cycle, measurement results may differ depending on the timing of the measurement. However, the present results suggest that a method to evaluate the side-to-side difference without considering the menstrual cycle may still provide reasonable results.

Furthermore, this study found differences in the magnitude of joint laxity between the reconstructed and contralateral sides. AKL was significantly higher on the reconstructed side than on the contralateral side. AKL was reported to increase after surgery using ST tendon grafts [36]. The AKL on the reconstructed side may therefore be greater than the AKL on the contralateral side. GR was lower on the reconstructed side than on the contralateral side, in contrast to the AKL. Patients with ACL injury often have hyperextension of the knee joint [37], which is also a risk factor for ACL injury [7]. After ACL reconstruction surgery, the range of motion in knee extension is often more limited than that on the reconstructed side [38]. The present study showed the same result.

In addition, no changes in muscle stiffness (ST and BF) were seen during the menstrual cycle in this study. A previous study using MyotonPRO to measure muscle stiffness in the thigh in the early follicular, ovulatory, and luteal phases showed higher stiffness in the medial vastus muscle and ST in the ovulatory phase than in the luteal phase [39]. On the other hand, another study found no difference in the resting muscle stiffness of the medial gastrocnemius and tibialis anterior muscles between early follicular and ovulatory phases using MyotonPRO [40], and muscle stiffness during the menstrual cycle remains controversial. The present study found no difference in E2 concentration between cycles, which may have explained why no change in muscle stiffness was identified.

This study had several limitations. First, the E2 concentration did not differ significantly between the early follicular and ovulatory phases of the menstrual cycle. Previous research examining the relationship between the menstrual cycle and musculoskeletal injury has shown that the incidence of injury is higher in those with irregular menstruation compared to those with normal menstruation [41]. This study recruited subjects with cycles between 25 and 38 days, but we could not be absolutely certain that the subjects would have normal menstrual cycles throughout the year. Second, the subjects also had different surgeons, different lengths of time since surgery, and different rehabilitation protocols. As a result, changes in joint laxity may not have been caused by the E2 concentration. Considering the postoperative remodeling of the ligaments [32], performing the measurements at a similar length of time postoperatively is most desirable. Although the duration of time since surgery may affect the results, all subjects in this study had undergone surgery over 1 year earlier, the composition of the reconstructed ligament was unlikely to change, and all participants had undergone the procedure using ST tendon. Third, the sample size was too small to draw solid conclusions because subjects were limited to those who had normal menstruation at the time of the experiment, had a history of unilateral ACL injury, and had undergone reconstruction of the ACL using ST tendon. A large sample size is needed to conclude the fluctuation of joint laxity on the reconstructed side during the menstrual cycle. Fourth, four of the nine subjects were involved in competitive sports club activities including baseball and volleyball. However, due to the coronavirus disease 2019 pandemic, subject activities were restricted, and they practiced only 1 h each day. Therefore, activity levels could not be completely standardized among all subjects, but subjects at somewhat the same activity level could be measured. Fifth, in terms of the characteristics of the MyotonPRO, the device analyzes muscle properties from the skin surface. For this reason, measuring the stiffness of the deeper layers of muscle is impossible, and the results of this study should thus be noted as representing superficial muscle stiffness. Finally, since the hormonal profile at the time of this study may not be similar to that at the time of ACL injury, correlating the occurrence of ACL injury with cyclic variation in joint laxity is not possible. Future prospective studies are needed to simultaneously follow the menstrual cycle and joint laxity in athletes.

## 5. Conclusions

In this study, AKL on the reconstructed and contralateral sides showed different patterns during the menstrual cycle (AKL on the reconstructed side was lower in the ovulatory phase than in the early follicular phase, while AKL on the contralateral side did not differ significantly between early follicular and ovulatory phases). This indicated that the reconstructed ligament and native ACL may respond differently to E2. However, no significant difference in AKL side-to-side difference was seen between the early follicular and ovulatory phases, suggesting a lower necessity of considering the menstrual cycle in clinical settings.

## Figures and Tables

**Figure 1 ijerph-20-02277-f001:**
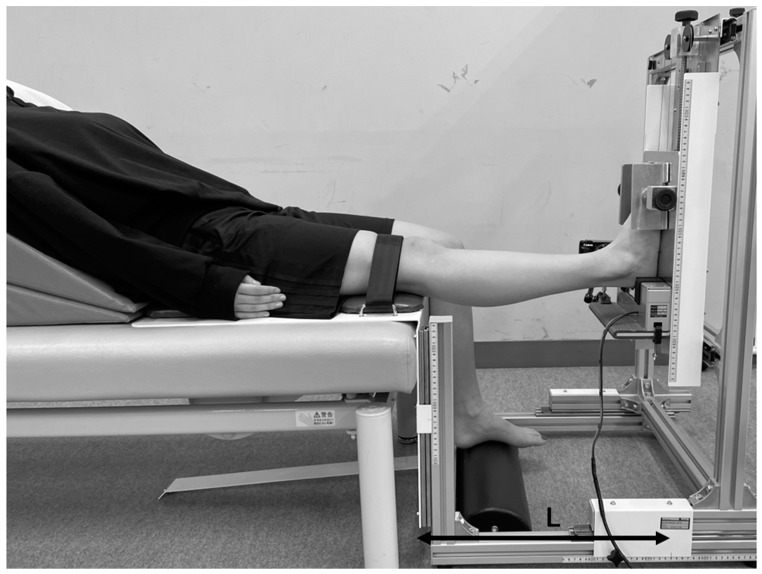
Position for measurement of genu recurvatum. The subject sat on the hyperextension apparatus in a relaxed position on the backrest cushion. Hip flexion was approximately 50° and the upper limbs were kept at the sides of the body. Length from the center of the knee to heel (L) and foot elevation values (H) were recorded. Maximum passive extension angle of the knee joint was later calculated.

**Figure 2 ijerph-20-02277-f002:**
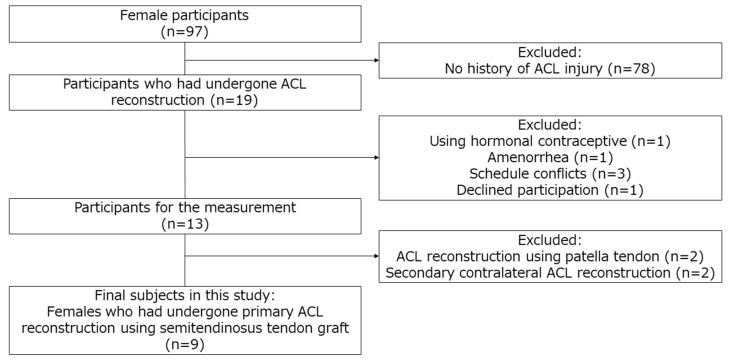
Flowchart for subject selection. ACL, anterior cruciate ligament.

**Table 1 ijerph-20-02277-t001:** Demographic data of the subject.

Height (cm)	160.4 ± 5.7
Weight (kg)	55.3 ± 5.0
BMI	21.5 ± 1.5
Age (year)	20.4 ± 1.5
Age at injury (year)	15.3 ± 1.76
Time from injury to surgery (month)	2.7 ± 1.5
IKDC subjective evaluation score	91.1 ± 7.0
Tegner Activity Scale before surgery	6.7 ± 0.9
Tegner Activity Scale after surgery	6.1 ± 1.1

Average ± standard deviation; IKDC: International Knee Documentation Committee.

**Table 2 ijerph-20-02277-t002:** E2 concentration, AKL side-to-side difference.

	Early Follicular Phase	Ovulatory Phase	*p*	*r*
E2 concentration (pg/mm)	1.2 [0.9–1.6]	1.5 [1.1–1.8]	0.374	0.333
AKL side-to-side difference (mm)	3.3 [2.9–4.4]	3.5 [1.0–4.0]	0.051	0.733

Median [interquartile range]; E2: Estradiol; AKL: anterior knee laxity; *p*: *p*-value using Wilcoxon signed-rank test; *r*: effect size calculated using Rank Biserial Correlation.

**Table 3 ijerph-20-02277-t003:** Comparison of the early follicular and the ovulatory phases.

	Reconstructed Side				Contralateral Side			
	Early Follicular	Ovulatory	*p*	*r*	Early Follicular	Ovulatory	*p*	*r*
AKL (mm)	9.4 [7.3–9.7]	8.3 [5.9–9.3]	0.044 *	0.756	5.3 [4.7–6.9]	4.8 [4.4–7.5]	0.515	0.244
GR (°)	7.7 [5.6–9.1]	7.5 [6.4–9.9]	0.086	0.644	9.7 [7.1–12.0]	10.2 [7.9–12.0]	0.173	0.511
ST stiffness (N/m)	194.0 [173.5–226.8]	196.7 [181.3–228.8]	0.213	0.467	191.0 [169.0–238.1]	199.7 [166.1–225.5]	0.953	0.022
BF stiffness (N/m)	188.8 [178.3–222.6]	203.7 [191.3–224.3]	0.086	0.644	192.0 [186.4–229.5]	210.5 [183.9–232.9]	0.514	0.244

Median [interquartile range]; AKL: anterior knee laxity; GR: genu recurvatum; ST: semitendinosus; BF: biceps femoris long head; *p*: *p*-value using Wilcoxon signed-rank test; *: *p* < 0.05, a significant difference between the early follicular and the ovulatory phases; *r*: effect size calculated using Rank Biserial Correlation.

**Table 4 ijerph-20-02277-t004:** Comparison of reconstructed and contralateral sides.

	Early Follicular Phase				Ovulatory Phase			
	Reconstructed	Contralateral	*p*	*r*	Reconstructed	Contralateral	*p*	*r*
AKL (mm)	9.4 [7.3–9.7]	5.3 [4.7–6.9]	0.008 *	1.000	8.3 [5.9–9.3]	4.8 [4.4–7.5]	0.021 *	0.867
GR (°)	7.7 [5.6–9.1]	9.7 [7.1–12.0]	0.021 *	0.867	7.5 [6.4–9.9]	10.2 [7.9–12.0]	0.028 *	0.822
ST stiffness (N/m)	194.0 [173.5–226.8]	191.0 [169.0–238.1]	0.514	0.244	196.7 [181.3–228.8]	199.7 [166.1–225.5]	0.594	0.200
BF stiffness (N/m)	188.8 [178.3–222.6]	192.0 [186.4–229.5]	0.341	0.356	203.7 [191.3–224.3]	210.5 [183.9–232.9]	0.594	0.200

Median [interquartile range]; AKL: anterior knee laxity; GR: genu recurvatum; ST: semitendinosus; BF: biceps femoris long head; *p*: *p*-value using Wilcoxon signed-rank test; *: *p* < 0.05, a significant difference between the early follicular and the ovulatory phases; *r*: effect size calculated using Rank Biserial Correlation.

## Data Availability

The data supporting the study’s conclusions are accessible from the author.

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
