# Peer review of "Knee Laxity in the Menstrual Cycle after Anterior Cruciate Ligament Reconstruction: A Case Series"

_ijerph, 2023, doi:10.3390/ijerph20032277_

Round 1

Reviewer 1 Report

Dear authors,

Thank you for your efforts in your research. I think the research you have examined regarding women's menstrual cycle is very valuable in terms of literature. Although your findings may seem small, the result is important to show the effects of menstrual phases on ACLR, so after minor verifications I will give below, I think that your research is suitable for publication in ijerph journal.

-You have given the main hypothesis in your research, but if there are any of your secondary hypotheses that you find important, please add them.

-There are some grammatical errors in English, I recommend you to check it.

-You have written the limitations of your research completely and descriptively. However, I did not see anything related to the number of subjects. If the number of subjects does not pose a limitation, add your power analysis results to the method section for the number of subjects.

Yours sincerely

Author Response

Thank you for your feedback which helped us improve this paper. Please see the attachment.

Reviewer 2 Report

This manuscript represents a study with the aim to compare anterior knee laxity (AKL), genu recurvatum (GR), and muscle stiffness between the reconstructed and contralateral sides in women who underwent unilateral anterior cruciate ligament (ACL) reconstruction during the follicular and ovulatory phases. The study should be of great interest to readers. These are my comments and suggestions:

Abstract:

Please, explain all the abbreviations used in the abstract. I would advise to add quantitative data in the abstract (especially where difference was found).

Introduction:

Nicely written, except I would advise to add references into this sentence: "Previous studies that examined joint laxity changes..."

Methods:

Did you make power analysis (regarding the study sample)? Were your data normally distributed for parametric tests?

Results:

I would advise to add the table for demographic data of the participants, including time from the surgery. It would be useful to calculate effect size.

Discussion:

Nicely written, however I am worried that your study is underpowered. Please, elaborate on this in the Discussion.

Author Response

Thank you for your feedback which helped us to improve this paper. Please see the attachment.

Reviewer 3 Report

Dear Authors

First of all, thank you for submitting to IJERPH.

Implants used for ACL reconstruction are fibrous tissue.

Depending on the menstrual cycle, the laxity of the ACL is very minute and cannot be measured with the equipment.

If laxity is confirmed enough to be measured by the equipment, it means that the operation has failed, and it is not affected by the menstrual cycle.

Besides, non-significant results are all too natural. It is by no means a new discovery. Even if the author discovered laxity, it cannot be said to have academic value.

It is very cautious to explain any result with data from only nine patients.

The results in Table 1 alone provide too little information to be published as a paper.

Unfortunately, I would like to tell you that the publication of this study is difficult.

Author Response

(The authors gave the same response as above.)

Round 2

Reviewer 3 Report

Accept in present form